# A Narrative Review of the Government Wheelchair Provision System in India

**DOI:** 10.3390/ijerph18105109

**Published:** 2021-05-12

**Authors:** Shivani Gupta, Agnes Meershoek, Luc De Witte

**Affiliations:** 1Centre for Inclusive Policy, Puttaparthi 515134, India; 2Department of Health, Ethics and Society, University of Maastricht, 6229 HA Maastricht, The Netherlands; a.meershoek@maastritchtuniversity.nl; 3Health Services Research, The Innovation Centre, The University of Sheffield, Sheffield S1 4DP, UK; l.p.dewitte@sheffield.ac.uk

**Keywords:** wheelchair provision, India, wheelchair services, procurement, right to mobility, rural areas

## Abstract

Background and aim: India has had a wheelchair-delivery system in place for several years but its impact on users is inadequate. Therefore, this research reviews the system to examine how the right to personal mobility can be served better. Method: this paper undertakes a narrative review of the existing government-aided wheelchair provision system from the perspectives of legislation and implementing agencies, both governmental and non-governmental, through document review and key informant interviews. Results: the results indicate that all steps of the government-funded wheelchair provision system are executed by the same system. Manufacture and supply take place nationally, but wheelchair services are largely absent. Moreover, the right to access mobility devices is not upheld for all users. Conclusion: the established government-aided wheelchair provision system is inadequate in terms of coverage, design, production, supply, and wheelchair services. Therefore, there is a need to reconsider the system by increasing its coverage and creating partnerships between the government, non-governmental agencies, and private agencies to improve access.

## 1. Introduction

For some persons with disabilities, having access to mobility devices is a pre-condition to living independently and freely accessing all rights, such as education, employment, and political participation Article 20 on personal mobility of the United Nations Convention on the Rights of Persons with Disabilities (UNCRPD) requires the State to ensure that persons with disabilities have independence in their mobility in a manner and at a time of their choice [1]. The UNCRPD entitles all persons with disabilities to access affordable assistive devices as a means to ensure their full and equal enjoyment of all human rights and fundamental freedoms. Countries that have ratified the UNCRPD are obliged to ensure this [2]. 

The Government of India has taken significant steps to make assistive devices—including mobility devices—available to persons with disabilities. First, mobility devices have been interpreted as a constitutional claim and a pre-condition for the citizens with disabilities to enjoy their right of freedom of movement [3]. Second, the government has invested in a robust system to make mobility devices available to persons with disabilities through the Assistance to Disabled Persons (ADIP) scheme, for purchasing/fitting of aids/appliances [3]. Finally, it has established Artificial Limbs Manufacturing Corporation of India (ALIMCO), the largest manufacturer of mobility and rehabilitation aids in South Asia, to provide affordable assistive products [3]. 

Mobility devices in India are largely available to persons with disabilities through the government’s ADIP scheme, through philanthropic or non-governmental organizations as donations, or as direct purchases by the users from the open market. According to available data, only 20% of persons with disabilities requiring such devices have been advised to acquire one, of which, only 16% have a device. Only one-fifth of them acquired their device through a government scheme while almost two-thirds purchased the device themselves ([4], p. 48). The state-wise variation in the percentage of users who received their device from the government [4] reflects in the state-wise differences in the implementation of the government scheme. Other reasons for the poor reach of the scheme are the lack of awareness of the scheme among state governments and persons with disabilities themselves [4,5].

The reach and impact of the scheme being inadequate, there is a need to probe beyond the reasons already documented, such as the lack of awareness. This research investigates how access to personal mobility devices can be improved for persons with disabilities in India. This research focuses on wheelchair provision under the scheme and examines the wheelchair provision system of the government. The reason for using wheelchairs as an example of assistive devices is that they are commonly used mobility devices that are expected to not only increase mobility, but also improve the health of users, and increase their socio–economic wellbeing and participation in community activities [6,7,8]. Moreover, India recognizes the WHO guidelines on the provision of manual wheelchairs in less-resourced settings [6] (hereinafter referred to as the WHO Guidelines), which are used to describe and analyze the wheelchair provision in this report. 

## 2. About the ADIP Scheme

Since 1981, the Indian government has been running the ADIP scheme for the provision of assistive aids and appliances—including wheelchairs—to eligible persons with disabilities. It is the largest assistive technology provision system in the country. The scheme has been revised several times, with the latest revision being made in 2017 [9]. The ADIP scheme has two parts. The first part is funded and implemented directly by the Ministry of Social Justice and Empowerment (MSJE). It is implemented through government and non-governmental organizations (NGOs). The other part is a cost-sharing scheme with the MSJE contributing 60% funding and 40% funds being contributed by the Ministry of Human Resource Development. This part of the scheme aims to provide aids and appliances to children and schools. 

### Eligibility and Quantum of Support

Under the ADIP scheme, any person with at least 40% disability and a total monthly household income of less than INR 15,000 (approximately USD 203) receives a free assistive device, including wheelchairs or tricycles. Those with a monthly household income between INR 15,000 and INR 20,000 (approximately USD 270) receive a 50% subsidy [9]. Eligible persons can receive an aid that costs a maximum of INR 10,000 (USD 135) from the list of aids approved by the scheme and can reapply for a new manual wheelchair after three years. 

## 3. Methodology

This narrative review adopts a qualitative research methodology. A narrative review was selected as it addresses the question in a broader scope and offers an indication of the steps to take while developing a policy. It also offers a critical review that leads to new insights for further research [7]. A narrative review also enables combining the literature review with primary research methods as described below [7].

### 3.1. Research Design

The research was undertaken in three steps that included a document review, interviewing key informants, and field observations of persons with disabilities. These steps were adopted to obtain information from different perspectives, namely, the efforts of the government, the implementation of the scheme and the impact of wheelchair provision on the ground. The three steps are elaborated below:

#### 3.1.1. Government Efforts

A document review of the existing ADIP scheme and the process of wheelchair provision under the scheme was undertaken. The documents reviewed were from three government websites. These websites were of the MSJE, which is the nodal ministry responsible for ensuring disability rights, the Department of Empowerment of Persons with Disabilities (DEPD) that operates under the MSJE, and ALIMCO, which is a government enterprise operating under the MSJE and the key manufacturer of wheelchairs for the scheme. 

Key official documents reviewed include:ADIP scheme guidelines [9];Annual Report of the DEPW 2020–2021 [8];47th Annual Report 2019–2020 of ALIMCO [10];ALIMCO product list on ALIMCO website [11];ALIMCO Quality Policy [12];First country report of India on the status of persons with disabilities submitted to the committee for the rights of persons with disabilities [3];Common guidelines for centrally sponsored schemes through NGOs [13].

#### 3.1.2. Implementation of Wheelchair Services

Information on the implementation of wheelchair services was undertaking through telephone interviews with two key informants, one of whom is working with the government-implementing agency and the other in an NGO that was an implementing agency for the scheme in the past. Both interlocutors are professionals engaged in wheelchair services with the first-hand experience of working with users. They were recommended by the organizations of persons with disabilities (OPD).

Open-ended interviews were undertaken based on questions derived from the WHO Guidelines, focusing on the implementation of the scheme. The questions asked were related to the process they followed for providing a wheelchair to users, elaboration on the various aspects of wheelchair service and the challenges they faced. 

Information collected from the document review complemented the data in the analysis.

#### 3.1.3. Impact of Wheelchair Provision on the Ground

The impact of wheelchair provision on the ground was undertaken in two ways. First, observations of ten persons with mobility impairments from three villages in the Anantapur district of Andhra Pradesh were undertaken. All of these persons required wheelchairs for mobility and were eligible to receive a free wheelchair as per the ADIP scheme. Observations were made in their homes, focusing on their mobility within different sections of their home, for which prior consent was taken. 

Second, a key informant who works with persons with disabilities on the ground was interviewed. This informant heads an OPD working with beneficiaries of the ADIP scheme. He is also a user of a wheelchair service. 

### 3.2. Scope of the Research

The ADIP scheme has two parts as described above—one that addresses persons with disabilities in the community and the other that is linked to the Sarva Shiksha Abhiyan (SSA), a government program for the universalization of elementary education in the country that addresses children with disabilities who are a part of SSA. This research focuses only on wheelchair provision to persons with disabilities in the community, and does not explore the branch of the scheme associated with the SSA.

### 3.3. Analysis

The analysis is built on the WHO Guidelines, which elaborate on four different steps in wheelchair provision: design, production, supply, and wheelchair services [6]. Data collected under each of these steps were analyzed for government efforts, implementation of wheelchair services, and the impact of wheelchair provision on the ground. 

## 4. Results

The section on government efforts addresses wheelchair design, production, and supply, as these are controlled by them. This part of the analysis is based on the document review. Implementation of wheelchair services is based on key-informant interviews of personnel from government and non-governmental organizations. The impact of wheelchair provision on the ground was collected through observations and by talking to a key informant. 

### 4.1. Government Efforts

The government recognizes its role in wheelchair provision and provides free wheelchairs to persons with disabilities below a specified income bracket through the ADIP scheme. This scheme enables access to wheelchairs to a large number of people annually. In seven years until 2020, the ADIP scheme utilized approximately INR 10.8 billion (USD 0.14 billion), benefitting 1.8 million beneficiaries of various assistive devices. These devices have mostly been provided through the 9460 camps organized in this period. Disaggregated data based on the number of persons who were given wheelchairs, is not available. 

There are concrete measures taken by the government for producing wheelchairs locally using local raw materials. ALIMCO manufacturers wheelchairs [3], and has a monopoly on the production and supply of wheelchairs under the ADIP scheme [10]. This eliminates the need for the government to purchase them from the open market or import them.

ALIMCO has four models of wheelchairs. These include two models of adult folding manual wheelchairs (of which, one is with a detachable arm), a model of adult rigid wheelchair and a pediatric folding wheelchair with fixed armrests [11]. The wheelchairs offered are elementary in design, without options for fitting the chair to the body size of the user or providing postural support.

National standards apply to the wheelchairs manufactured by ALIMCO. ALIMCO adheres to these standards and has a quality policy committing to produce and supply quality assistive devices to persons with disabilities and older persons at a reasonable price [12]. All ALIMCO wheelchairs are based on Indian Standard (IS): 6571 of 1991 and IS: 7454 (1991) revised in 1995. These standards have not been updated, nor are they mandatory for private wheelchair manufacturers. 

The supply of wheelchairs is undertaken by implementing agencies funded by the government for the purchase, fabrication, and distribution of aids and appliances. Implementing agencies selected under the ADIP scheme must have professionally trained staff to deliver wheelchair service and the organization must have the machinery for the fabrication, fitting, and maintenance of the aid [8]. 

The implementing agencies largely include government organizations such as the national institutions working under the DEPW, the Composite Regional Centers (CRC), District Disability Rehabilitation Centers (DDRC), and ALIMCO. Hospitals, NGOs, and voluntary organizations can also function as implementing agencies. Implementing agencies procure the wheelchairs from ALIMCO. 

To summarize, the Indian government has an elaborate system for wheelchair provision under the ADIP scheme. Persons with disabilities who earn less than a specified monthly income are entitled to receive a free wheelchair that can be replaced after three years. However, there is no system to repair or replace parts. An earmarked annual budget is allocated to the scheme. ALIMCO, a government enterprise, has been set up for the production and supply of wheelchairs. Therefore, the money spent on procuring wheelchairs is plowed back into the system. However, wheelchair design options are limited and there is limited scope for customization and fitting. These wheelchairs are designed based on national standards but the standards have not been regularly updated. Moreover, the standards do not apply to private manufacturers. Wheelchairs are supplied and wheelchair services are provided across the country through implementing agencies.

### 4.2. Implementing Organizations

The organizations implementing the ADIP scheme may be government-owned or NGOs. According to key respondent B, at one time NGOs were more involved in the implementation of the scheme, but of late, the role of governmental organizations has increased. She said:
“The government system seems to be changing slowly and now, largely it is the government agencies that receive funds for implementing the scheme. As a result, our organization has stopped requesting funds, as we think it is a waste of time.”

Her view reflects in the annual report of the ministry that provides details of organizations that have received funds for ADIP [8]. The selection criteria for NGOs are stringent. They require a recommendation from the state government and a third-party evaluation of the organization, amongst other requirements [13]. 

These implementing agencies are also responsible for fitting, following up, and providing support for the maintenance of wheelchairs [9]. However, the budgetary allocations made by the government do not reflect a budget heading for maintenance or follow-up [8]. 

According to key informant A, the scheme is implemented through two systems of distribution; headquarter and camp,
“There are two main ways of providing devices. One is the headquarter distribution, under which the user comes in person to the national institute or an implementing agency. The eligibility of the user is verified before they are assessed, fitted and provided with the assistive device. Camp distribution is the other route. Here, information about eligible persons identified through local-level mechanisms (such as the panchayat and block-level rehabilitation workers) is collected by the District Disability Welfare Officer (DDWO). Once the officer has information about a certain number of persons with disabilities requiring assistive devices, the national institute is invited to hold a camp in the district. Potential users are informed in advance and invited for assessment. All assessments are collected and the required devices procured. The team returns for a fitting camp where assessed users are fitted with the device.” 

In addition to camps funded by the central government and organized based on the proposal of the DDWO, there are special camps organized based on the proposal of political dignitaries, such as Members of Parliament (MP) and Members of the Legislative Assembly (MLA), using funds sanctioned to them, and in collaboration with ALIMCO. Camps are also organized by ALIMCO in partnership with other central public sector undertakings as a part of their corporate social responsibility initiatives. In the financial year 2019–2020, ALIMCO organized 53 such special distribution camps. Such camps see the presence of high-level political figures during the distribution. Five such mega distribution camps were attended by the Prime Minister of India. Some of these special camps have gained an entry to the Guinness Book of World Records for the largest distribution of a disability device [8].

Camp distribution is the most significant method of delivering wheelchairs to persons with disabilities. Every year, over a thousand such camps are organized throughout the country where persons with disabilities are assessed and assistive devices distributed. In 2019–2020, over INR 2.5 billion (approximately USD 34 million) was spent in organizing 1156 camps where over 150,000 persons with disabilities were provided assistive devices [7].

Since fitting, training, maintenance, and follow-up are responsibilities of the implementing agencies, when asked how activities were undertaken, key informant A said,
“At the fitting camp, group training may be provided to the users on the use and care of the device they have been provided with. No other maintenance or follow-up is provided. However, since the national institutes have information on the beneficiaries, they contact the recipients after three years for a replacement. The beneficiaries may not always return for the replacement.”

Key informant B verified that,
“Unlike in the case of other devices, no fitting or assessment is done for wheelchairs; they are just distributed. The WHO Guidelines on the provision of manual wheelchairs (2008) were rolled out in India but are not followed. The ADIP scheme implementers were provided with a two-day orientation on the WHO Guidelines but it never became a part of the system. So, wheelchairs and tricycles are distributed en masse, without a cushion.” 

She highlighted that,
“The practitioners under the ADIP scheme are often not trained to undertake wheelchair fittings and measurements, and even if they were, it would not be useful as there are no wheelchair options available. Also, the number of people they cater to in one camp is large; they hardly have time for individual attention.” 

Encapsulating the findings from the implementing agencies highlights that these agencies are funded by the government and most of the wheelchair distribution is done through camps. They are also expected to provide wheelchair services, though, in effect, they are unable to provide any—fitting, training, maintenance, or follow-up. Perhaps this is because there are no funds earmarked for services and the number of users at a camp is too large for individual attention. Therefore, wheelchairs are literally ‘distributed’ without any services being offered. Moreover, these camps are often used to gain political mileage. To be a recipient at such camps, a user must submit their eligibility documents to the DDWO. Upon receiving a sufficient number of applications, the DDWO sends a proposal to the implementing government agencies for a camp. The users are invited to the camp for assessment. Subsequently, the implementing agency staff places an order for procurement. Once the wheelchairs are procured, a distribution camp is organized. Thereafter, there is no follow-up for three years. Finally, while India has rolled out the WHO Guidelines of 2008 and related training has been provided to practitioners, the guidelines are not followed, especially for wheelchair services. 

### 4.3. On-the-Ground Impact of Wheelchair Provision

The ADIP scheme details the eligibility, quantum of support available, and role of the implementing agencies. This section presents the impact of the scheme on the lives of persons with disabilities.

Field observations show that a small number of persons who required a wheelchair, had one, and these were largely young men with disabilities. The older persons with disabilities were not aware of the benefits of having a wheelchair, and the young women with disabilities had a wheelchair at some point, but when it broke, they did not receive a new one, as they did not see it as being of use. Key respondent B suggested,
“While under the ADIP scheme a large number of devices are distributed, there is no government data available on how many persons continue to use these wheelchairs and tricycles.” 

One of the reasons for the persons with disabilities not receiving a wheelchair was the difficulty in applying for it. Key informant C mentioned that,
“Persons with disabilities who are eligible to receive a wheelchair under the scheme often found it difficult to furnish the documentation required for getting the device, which included copies of an income certificate, disability certificate, Aadhaar card (national identity card) and two passport-size photographs.” 

For persons with disabilities, putting together the documents and submitting these to the authorities to participate in a camp is not easy, especially because they are dependent on their families to contribute to the effort. Moreover, they need to undertake three trips; to submit their documents, to attend the evaluation camp, and to visit the fitting camp. According to key informant C, there are shortcomings in the way the scheme is implemented. He said,
“The Composite Rehabilitation Center in Srinagar, Jammu & Kashmir is the implementing agency but they did not have a director for the past twenty years. As a result, no outreach or awareness-raising programs were conducted, and people had to wait for long to get the mobility device.” 

He further suggested that,
“There is a long delay in actually getting the mobility device after making a request and submitting the eligibility documents. The delay is more than a year, at times.”

Another shortcoming that persons with disabilities see is in the inappropriate design of the wheelchair for the purpose it is meant to serve. Observations from the field revealed that since the persons with disabilities were not able to transfer themselves, they had to be lifted onto the wheelchair. Most wheelchairs—being of the standard design without detachable arms—lifting and transferring them onto the wheelchair was a strenuous task for family support providers. This was all the more difficult in rural India where most activities such as sitting, sleeping and eating are performed on the floor, requiring persons with disabilities to be physically lifted from the floor to be placed on a wheelchair. 

Moreover, most persons with disabilities were unable to self-maneuver in their external environments at times because of inaccessible pathways, their functional limitations, and lack of strength to propel the wheelchair. They needed support to push their wheelchair, limiting its use to the times when someone was available.

Finally, an important aspect the users expressed was the poor durability of the device, especially in the absence of maintenance and follow-up by the implementing agencies. According to key respondent C,
“The wheelchairs that people get through the ADIP scheme are like disposable products. They break in about three months. No customization is possible and the persons with disabilities get either an adult chair or a pediatric chair.”

Furthermore, the wheelchairs offered are only meant for external mobility, especially in rural India. Observations showed that all of these persons had to be physically lifted or had to make do with dragging themselves within the house. 

Summing up the user experience of wheelchair services suggests that most persons with disabilities lacked the knowledge, or found it hard to submit eligibility documents or participate in the camps. Even if they did participate at times, they got the equipment after a year, especially in places where the implementing agencies may not have a head or are understaffed. The users are unhappy with the durability of the wheelchairs and felt these did not last three years after which they could be replaced. Moreover, the wheelchair they received was not always suited to them or their environment. This could not be used indoors and it was difficult to transfer themselves into it. The wheelchairs could not be adjusted to meet individual requirements, as only basic models were available.

## 5. Discussion

This research reviews the government system of wheelchair provision in India. It shows that the government has an elaborate system implemented by the ADIP scheme, under which, a large number of assistive devices—including wheelchairs—are distributed every year. The robust government system of wheelchair provision is backed by a policy and wheelchairs are manufactured locally, based on national standards. Wheelchair distribution is handled by the implementing organizations, which are required to offer wheelchair services as well. The entire system is funded by the government. We discuss this further, based on the analysis.

### 5.1. Wheelchair Design

This research suggests that the aspects of wheelchair design suggested by the WHO Guidelines are not addressed by the ADIP scheme. None of the wheelchair designs that ALIMCO manufactures has postural support and an attached seating cushion. As a result, not only are some persons with disabilities unable to use these chairs but also their health and safety are compromised. Further, the users point to the non-durability of the wheelchair, and in the absence of maintenance services, only a few persons with disabilities—especially in rural areas—have a wheelchair. As a result, they hardly go out of their homes and wheelchairs do not enhance their participation in community matters. Moreover, the wheelchair design is unsuitable for mobility within the home, which further reduced its usefulness to users. However, the inadequacy of the wheelchairs for in-home mobility in India is not limited to the ADIP scheme wheelchairs. It is a common challenge highlighted in other literature as well [14,15,16]. 

### 5.2. Wheelchair Production and Supply

The government has complete control over the production and supply of wheelchairs distributed under the scheme through ALIMCO’s monopoly. Procuring only from ALIMCO can be considered a strategy for maximizing the social benefit from the resources available, by plowing the procurement expenses back into the government system. Consolidation of government procurement has several benefits to the scheme, such as the devices being available at a reasonable price and easier and seamless supply. 

However, the drawback of such control from the user’s perspective is the limited choice of wheelchairs available to them. Often, persons with disabilities do not use a wheelchair since it is unsuited to their environment and there is a high percentage of rejection of wheelchairs. Reports have suggested that users sell the wheelchair as scrap [15,17], thus compromising the primary aim of the scheme. 

The lack of wheelchair options is because ALIMCO offers only three basic models of wheelchairs for adults with disabilities, provided without a cushion. The notion of ‘one size fits all’ with limited customization adopted by the scheme renders a wheelchair prescription redundant. Not having a prescription automatically results in the user having an inappropriate wheelchair that is not properly fitted. The lack of wheelchair options with postural support makes them unusable by some persons with disabilities. Furthermore, not having a cushion may add to bodily complications such as a pressure sore, making the chairs unsafe. The scheme needs to provide more options of wheelchairs that can be customized to meet the postural and mobility requirements of users.

Complete control on production and supply also restricts the development of private manufacturers, thereby reducing the availability of quality wheelchairs at affordable prices in the open market. This impacts persons with disabilities who procure wheelchairs from the open market.

### 5.3. Wheelchair Services

According to WHO [6], the key functions of wheelchair services include assessment, provision, training, support, and referral. The scheme guidelines make the implementing agencies responsible for this [9]. However, on the ground, there are no services available to users. Existing research suggests that users may be less dissatisfied with the wheelchair itself, than with the absence of wheelchair services, especially training in wheelchair skills and its maintenance [18].

One reason for this could be how outreach camps are organized. India is a large country with 70% of persons with disabilities living in rural areas that lack trained practitioners. Therefore, camps may be an effective channel to reach this population. However, the way camps are organized needs reviewing. Often, camps are one-off events, also used to gain political mileage, making wheelchair distribution a charitable act rather than a right. Given a large number of beneficiaries at such camps, it is difficult to provide support services. Being one-off events, there are no follow-ups or maintenance services provided. As a result, the impact on the ground is not commensurate with the efforts put in.

It may also be highlighted that over 50% of households in the country qualify under the income criteria set for the ADIP scheme [19] and, therefore, the eligibility of the scheme covers a majority of persons with disabilities. However, for persons with disabilities above the income threshold, no efforts are made to address their right to mobility and they must buy wheelchairs from an unregulated open market [20]. Addressing the mobility of only a section of persons with disabilities in the country can be seen as being discriminatory towards others. 

Therefore, and because India has ratified the UNCRPD, there is a need to investigate the forty-year-old wheelchair provision system and the rights of all persons with disabilities must be addressed. This does not imply providing free wheelchairs to everyone, rather, taking alternative measures to increase access to wheelchair services for all. This can be done by creating alternative systems for funding the purchase and repair of wheelchairs, introducing wheelchair insurance, improving the availability of reasonably-priced quality wheelchairs in the open market, making wheelchair services a part of the Universal Health Coverage so that these can be provided equally in urban and rural areas with trained community-level workers, and adopting the WHO Priority assistive products list so that options for postural support and improved mobility are available. In rural areas, efforts are required to improve the indoor mobility of persons with disabilities through better-designed homes, and mobility devices suited to the rural habitat, as suggested in other literature [14]. 

## 6. Limitations

The analysis of the ADIP scheme implementation in this brief report is limited to two interlocutors, ten observations, and select government documents. A more extensive study representing different regions of the country, rural and urban, including interviews with government officials, implementing agencies, ALIMCO and private wheelchair manufacturers may give more nuanced details of the scheme and the challenges, which could account for potentially different implementation outcomes.

Further, this report focuses only on wheelchair provision while there are several other assistive devices provided under the ADIP scheme. Research that investigates the provision of all such products may bring out other aspects of the scheme not covered by this report. 

## 7. Conclusions

The wheelchair provision under the ADIP scheme fulfills several aspects of wheelchair provision described in the WHO guidelines. The present system demonstrates the country’s capacity to be self-sustained in wheelchair provision. However, a closer investigation reveals the inadequate reach and impact of the scheme on the lives of persons with disabilities. While the ADIP scheme has a holistic conceptualization addressing all aspects of wheelchair provision, the efforts are inadequate for a large country such as India. On the one hand, there are inadequacies in wheelchair design, shortcomings in implementation, and an absence of wheelchair services, while on the other, exists an insufficiency in addressing the right to mobility of all citizens. Moreover, the monopoly of ALIMCO in the production and supply of wheelchairs reduces the choices people have for type and customization. This influences the affordability and quality of wheelchairs available in the open market from where the majority of the users procure their wheelchairs. 

Therefore, there is a need to address wheelchair provision in a broader context by consolidating the demand of all users, including those eligible for government-aided wheelchairs, and those who are not. Production, supply, and wheelchair services to meet this consolidated demand would require collaboration between government agencies, non-governmental organizations, private manufacturers, suppliers of wheelchairs, and organizations of persons with disabilities. Such collaborations would not only increase the choice of wheelchairs and wheelchair services available in the market, but also improve the quality and affordability of these devices.

## Data Availability

The data used in this research is available to be shared if required.

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
