# Peer review of "A Narrative Review of the Government Wheelchair Provision System in India"

_ijerph, 2021, doi:10.3390/ijerph18105109_

Round 1
Reviewer 1 Report
The authors reviewed the wheelchair provision system proposed by the Indian government. I had some concerns:
1. Authors should specify what type of review it is? Is it explanatory or narrative or systematic review? At first look, I wondered that it could be the narrative type.
2. Methods are what kind of operations, which are performed to draw the results. In my opinion, the methods section seems a research background or literature review. Please mention what kind of analysis you are performed, and how these terms are searched. What kind of sources did you use?
3. It is hard to follow the work for readers when authors do not present clear sections and subsections of the manuscript. Please keep the sections that are the methods part, in the methods what are the subheadings including research scope and its design. Section 3 (or Results after 1. introduction and 2. Methods) should result in how authors come to the conclusions in results that mention there is a scarcity of wheel Chairs in the country. Clear connections between the contents are missing.
4. The discussion section is again literature on the limitations of connected health. It seems like too much literature was mentioned.
5. Line 465-466. The Indian government has near-automatic system control. I did not find any review on this issue necessity except too much information designing and working of wheelchairs.
Minor concern
Some grammatical, and language errors needed to be addressed.
Author Response
Response to REVIEWER 1
Dear reviewer,
We thank you for your feedback and an opportunity to improve the manuscript. We have addressed all the concerns raised by you. A brief elaboration of how we have addressed them is provided below in blue.
1. Authors should specify what type of review it is? Is it explanatory or narrative or systematic review? At first look, I wondered that it could be the narrative type.
We look at it as a narrative review as it is based on the review of the government documents bringing current and ground-level critical analysis of the system through interviews and observations.
We have therefore reflected the same in the title of the paper by adding it to the title and referencing it in the methodology.
2. Methods are what kind of operations, which are performed to draw the results. In my opinion, the methods section seems a research background or literature review. Please mention what kind of analysis you are performed, and how these terms are searched. What kind of sources did you use?
We have revised the entire methodology section highlighting the process we followed in collecting the data and analysis in greater detail.
- It is hard to follow the work for readers when authors do not present clear sections and subsections of the manuscript. Please keep the sections that are the methods part, in the methods what are the subheadings including research scope and its design. Section 3 (or Results after 1. introduction and 2. Methods) should result in how authors come to the conclusions in results that mention there is a scarcity of wheel Chairs in the country. Clear connections between the contents are missing.
We have created clear headings with subheadings in the methodology and in the analysis. Some new sub-headings have been created in the methodology to make comprehension easier.
- The discussion section is again literature on the limitations of connected health. It seems like too much literature was mentioned.
The discussion has been revised. Literature that was not contributing to the content has been removed and only selected literature that validates our discussion has been retained.
- Line 465-466. The Indian government has near-automatic system control. I did not find any review on this issue necessity except too much information designing and working of wheelchairs.
The discussion under the subheading 'wheelchair production and supply' where these lines feature, has been revised to address the control of the government on these activities. It reflects the benefits of the automatic system for the government and its impact on the users.
Some information about the design is still present in the section as these are indicators in the WHO Guidelines which are used as a framework for analyzing the data in this report.
Minor concern
Some grammatical, and language errors needed to be addressed.
Revised the entire report
Reviewer 2 Report
I read your article with great interest and have four broad suggestions. First, I think it would be useful to highlight the fact that your analysis of the program's implementation was limited to just two interlocutors and ten observations and suggest that a more extensive study that could arguably represent different regions of the nation, rural and urban, should be undertaken, to account for potentially different implementation outcomes. Secondly, I was struck that you contend that the chairs are of poor quality and all but uncustomizable, which would make them all but useless in practice for many individuals with CP, for example. But you do not indicate why this choice was made or why parts are not widely available either. Both of these policy choices would appear to compromise the program's efficacy from the start. Can you say more on these crucial concerns? Or call for necessary changes concerning them at the very least? Third, you suggest that rural infrastructures, both social and physical, are isolating for those you studied. That fact too limits the realization of the aims of the program, not to say the mobility rights of those it serves. What can be done and if it involves social (and, for that matter physical improvements too)change, what should the government be doing that it is not now undertaking, to secure the aims of this program beyond supplying deficient hardware? Finally, and in short, your article's findings suggest a program that far from realizing its purposes seems simply to be avoiding the real efforts necessary to realize those aims. The metaphor that comes to mind is the deficient supply of tiny Band Aids when the problem is a gaping wound in need both of surgical intervention and major long term bandaging as it heals. Is that a fair conclusion based on your investigation? If it is not, why not?
Author Response
Dear reviewer,
Thank you for your feedback and an opportunity to improve the manuscript. We have addressed all your concerns and a brief elaboration of what we did is provided below in blue.
- First, I think it would be useful to highlight the fact that your analysis of the program's implementation was limited to just two interlocutors and ten observations and suggest that a more extensive study that could arguably represent different regions of the nation, rural and urban, should be undertaken, to account for potentially different implementation outcomes.
We do realize the limited primary data we have for this research. That was the main reason for submitting the research as a brief report rather than a research paper.
We have added this in the limitations of the revised manuscript in the end.
Secondly, I was struck that you contend that the chairs are of poor quality and all but uncustomizable, which would make them all but useless in practice for many individuals with CP, for example. But you do not indicate why this choice was made or why parts are not widely available either. Both of these policy choices would appear to compromise the program's efficacy from the start. Can you say more on these crucial concerns? Or call for necessary changes concerning them at the very least? (lines 1184, and 1194)
The revised manuscript addresses this more elaborately under wheelchair design in the discussion. (lines 354 to 366). To draw more attention to the problem, aspects of unsuitable design are also now addressed in the next sub-section (lines 380 to 389). The issue of inadequate design and quality has been addressed in the conclusion as well.
Third, you suggest that rural infrastructures, both social and physical, are isolating for those you studied. That fact too limits the realization of the aims of the program, not to say the mobility rights of those it serves. What can be done and if it involves social (and, for that matter physical improvements too)change, what should the government be doing that it is not now undertaking, to secure the aims of this program beyond supplying deficient hardware?
We have added recommendations for the government in the revised manuscript (lines 417 to 430). The recommendations begin suggesting in light of the duty for the progressive realization of the right to mobility by taking a more universal approach in wheelchair provision that allows equal access in rural and urban areas and adopting the WHO Priority list of AT to ensure more options of wheelchairs.
Finally, and in short, your article's findings suggest a program that far from realizing its purposes seems simply to be avoiding the real efforts necessary to realize those aims. The metaphor that comes to mind is the deficient supply of tiny Band Aids when the problem is a gaping wound in need both of surgical intervention and major long term bandaging as it heals. Is that a fair conclusion based on your investigation? If it is not, why not?
The conclusion has been completely revised. The approach suggested is universal access to wheelchairs and services and market-driven response to it in collaboration with NGOs, private enterprises, and the organizations of persons with disabilities.
Round 2
Reviewer 1 Report
The authors have successfully addressed my concerns. Therefore, the manuscript may be accepted in this form.
Author Response
Dear reviewer,
Thank you.
Kind regards,
Shivani
Reviewer 2 Report
I have read this version and find it clearer than your last. I encourage you to revisit your abstract to do three things. First, I suggest you eliminate references to the "government"- a descriptor which does not make sense int his context. I suggest you refer instead and more precisely to relevant officials in this context. The "government" is not acting but those employees or contractors charged with relevant responsibilities surely are. Second and related, I suggest you revise what you say in the methods section of the abstract to say what you actually did vis-a-vis government actors. Last, I could not follow your conclusion section of your abstract- what is now offered is not sentence and in any case, not clear. Beyond those concerns, here are some additional specific comments. Irst, on line 67 you note you will use WHO guidelines to "conceptualize" wheel chair provision. But you do not do that. Instead, you use WHO guidance to compare as a rough and ready metric against which to gauge the government's program of wheel- chair provision. Second the section "About the ADIP Scheme" is a bit muddy regarding differences between provision of wheelchairs and other aids. Third, I suggest you provide a bit more information in line 111 and following concerning how and why you chose your two interlocutors. What roles did they occupy that made them good choices for your analytic purposes? Indeed, a simple table providing an overview of your data sources would be most welcome. Fourth, on page 9 you suggest that "most persons... do not have a wheelchair.. and do not feel the need for one" without providing data to found your claim. How do you know this and how do you know that they do not such support, properly understood? Indeed, how would many of those to who you very generally refer, know? Fifth, I suggest substituting a more precise term for "looked at" throughout. I assume you did more than this vague term suggests in your analysis throughout? Finally, as before I suggest you punch up your conclusions to highlight your key findings more clearly. Lines 448 and following are actually quite vague in this regard in light of what you found. Can you be more precise and thereby illuminate the issues that need to be addressed to serve users more effectively and equitably and also more concretely suggest the ways in which specific policy steps could address them?
Author Response
Dear Reviewer,
Thank you for your feedback. We have addressed all your comments. Please find a brief elaboration of what we did in blue
I have read this version and find it clearer than your last. I encourage you to revisit your abstract to do three things. First, I suggest you eliminate references to the "government"- a descriptor which does not make sense int his context. I suggest you refer instead and more precisely to relevant officials in this context. The "government" is not acting but those employees or contractors charged with relevant responsibilities surely are. Second and related, I suggest you revise what you say in the methods section of the abstract to say what you actually did vis-a-vis government actors. Last, I could not follow your conclusion section of your abstract- what is now offered is not sentence and in any case, not clear. Beyond those concerns, here are some additional specific comments.
We have revised the abstract incorporating all the comments
First, on line 67 you note you will use WHO guidelines to "conceptualize" wheel chair provision. But you do not do that. Instead, you use WHO guidance to compare as a rough and ready metric against which to gauge the government's program of wheel- chair provision.
We replaced “conceptualized with the word “conceptualize” with “describe and analyze”
Second the section "About the ADIP Scheme" is a bit muddy regarding differences between provision of wheelchairs and other aids.
We have revised textual changes to make it clearer.
Third, I suggest you provide a bit more information in line 111 and following concerning how and why you chose your two interlocutors. What roles did they occupy that made them good choices for your analytic purposes? Indeed, a simple table providing an overview of your data sources would be most welcome.
We have added details of the interlocutors who were both professional engaged in wheelchair services.
We have added a bulleted list of official documents reviewed
Fourth, on page 9 you suggest that "most persons... do not have a wheelchair.. and do not feel the need for one" without providing data to found your claim. How do you know this and how do you know that they do not such support, properly understood? Indeed, how would many of those to who you very generally refer, know?
We have deleted the line as it was not adding any additional information to the text.
Fifth, I suggest substituting a more precise term for "looked at" throughout. I assume you did more than this vague term suggests in your analysis throughout?
Reworded all occurrences of “looked at” and “looks at” with more appropriate words
Finally, as before I suggest you punch up your conclusions to highlight your key findings more clearly. Lines 448 and following are actually quite vague in this regard in light of what you found. Can you be more precise and thereby illuminate the issues that need to be addressed to serve users more effectively and equitably and also more concretely suggest the ways in which specific policy steps could address them?
We have rewritten the conclusion to provide a clearer picture of the issues and recommendations to address these.